# Sociodemographic characteristics as predictors of knowledge regarding mode of transmission of Lymphatic Filariasis among population of Nepal

**Ramesh Adhikari**[1,2☯], **Devaraj Acharya**[3☯]*, **Aakriti Wagle**[2]

**1** Mahendra Ratna Campus [Tribhuvan University], Kathmandu, Nepal, **2** Center for Research on Education, Health and Social Science (CREHSS), Kathmandu, Nepal, **3** Bhairahawa Multiple Campus [Tribhuvan University], Rupandehi, Nepal

☯ These authors contributed equally to this work.
* devaraj.acharya@bmc.tu.edu.np

**Data Availability Statement:** All relevant data are freely available and can be accessed through the DHS program (www.dhsprogram.com). The authors used the NDHS 2016 datasets for Nepal.

## Abstract

### Introduction

The majority of the districts of Nepal (63 out of 77) were detected as prevalent of Lymphatic Filariasis (LF), with an average prevalence of 13 percent ranging from less than one to 39 percent in Nepal. Despite people's ignorance about the LF, the government has a target to eliminate LF by 2020. The study aimed to assess the association between sociodemographic characteristics and knowledge of the mode of transmission of LF.

### Materials and methods

The study used a cross-sectional design. Secondary data from Nepal Demographic and Health Survey 2016 were analysed. Altogether, 11040 participants participated in the study. Sociodemographic characteristics were the independent variables, whereas household heads' knowledge of LF transmission was the dependent variable. Descriptive, bivariate, and multivariate analyses were performed to assess the association between sociodemographic characteristics and knowledge of the mode of transmission.

### Results

Findings showed that only 28 percent of household-heads correctly identified the mode of transmission of Lymphatic Filariasis in the study. Household head's age, sex, wealth status, place of residence in terms of areas, geo-belt and province, migration history, and household assets such as bed nets, Radio, TV were significantly associated with the knowledge of the mode of transmission of Lymphatic Filariasis. Variables: wealth status, sex, residence, eco-belts, possessing bed nets, and Radio appeared as significant predictors for knowledge of the mode of transmission of Lymphatic Filariasis. The richest to the poorest people had lower odds ranging from 0.22 to 0.53 for knowing the mode of transmission of Lymphatic Filariasis compared to the richest people (p = 0.001).

**Funding:** The authors received no specific funding for this work.

**Competing interests:** The authors have declared that no competing interests exist.

## Conclusion

The study identified the population groups with a low level of knowledge of modes of transmission of Lymphatic Filariasis. Thus, it can be inferred from the study that relevant programs need to focus on women, people residing in the mountains and Terai, and those belonging to the middle and poor wealth index. The study also emphasizes that information, education, and communication materials can effectively impart knowledge of Lymphatic Filariasis.

## Introduction

Lymphatic Filariasis (LF) is one of the neglected tropical diseases that still exists as a public health problem worldwide. LF was detected in 73 countries affecting 40 million populations with clinical manifestation [1]. World Health Organization estimated that about 85 percent of the total population required some interventions for neglected tropical diseases such as Dengue, Leishmaniosis, Scabies, Schistosomiasis, Lymphatic Filariasis in the South-East Asian Region in 2017 [2]. LF occupied the second position in terms of tropical disease burden and causes of disability worldwide, known as the disease of the poorest of the poor [3]. LF is caused by *Wuchereria bancrofti*, which is transmitted mostly from the bite of *Aedes mosquitoes*. LF causes morbidity and mortality both in human and live stocks. LF is said to be possible to eradicate, and the eradication program is going under the way [4]. Many countries have initiated mass drug administration (MDA) programs to reduce LF prevalence [5, 6]. Researchers found that LF negatively affects the individual's personal and social life, limiting the individuals' potentialities and occupational activities since dermatitis, lymphedema, enlargement of genitals, and swelling of legs may manifest due to LF [7].

Marginalized people who lived in poor sanitation and housing condition are more vulnerable to LF. Affected people suffered from social humiliation and disability with poor socioeconomic consequences [8]. People still do not have adequate knowledge about it. A study in South Jakarta shows that just two percent of primary health care workers had good knowledge about LF and MDA. However, after the health education intervention, 65 percent of the primary health care workers had good knowledge about it [9]. Similarly, another quasi-experimental study shows that good knowledge increased to 36 percent from eight percent in the control and experiment group after implantation of MANDIRI: a pocketbook program [10].

### The situation of Lymphatic Filariasis in Nepal

LF is one of the major public health problems in Nepal. Sixty-three out of 77 districts are identified as LF prevalent districts. Up to 39 percent prevalence of LF is recorded in some districts with an average of 13 percent of the total population [11, 12]. The magnitude of disease prevalence is detected more in poor, rural, and Terai than rich, urban, and hill areas, respectively; however, the valley and river basins in the hill are also noticed as having a high prevalence of the disease. LF is detected above 300 feet in Terai to 5800 feet in the hill from the sea level in Nepal. By 2020, the Government of Nepal [GoN] targets eliminating LF as a public health problem [12]. The GoN had initiated a mass drug administration program in 2003 in Parsa District [13] and have a target to reduce LF prevalence to less than one percent by 2020 [12], but migration and movement to the non-endemic area from endemic areas appears a challenge to meet the target of LF elimination [14].

Studies showed that knowledge about LF and its mode of transmission was associated with treatment-seeking behaviour that would support meeting the target to reduce its prevalence rate to less than one percent by 2020 [1]. Although Nepal Demographic and Health Survey (NDHS) 2016 shows that 92 percent of the population heard about LF, just 28 percent had good knowledge about its transmission [15]. LF is one of the neglected tropical diseases, and little attention has been paid to GoN, people, and researchers. Little research has been conducted and published about the disease, especially about knowledge of LF transmission. We could not find any study about the mode of transmission of LF concerning the people's socio-demographic and economic status. Therefore, the study aims to assess the knowledge of the mode of transmission of LF regarding the sociodemographic and wealth status of the respondents using national representative sample survey data.

## Materials and methods

NDHS 2016 was the national representative sample survey. It was the fifth survey of this type since 1996. It was conducted under the guidance of the Ministry of Health Nepal with the collaboration of New ERA with the technical assistance of International Finance Corporation and funded by United States Agency for International Development. The NDHS report has already been published and publicly available so all methodological issues can be obtained from the report [15].

## Population and sample

All households throughout the nation were the universe of the study, and multilevel sampling strategies were applied to make the results representative. The sampling frame was used based on the National Population and Housing Census 2011 conducted by the Central Bureau of Statistics [CBS]. Some political maps had been changed in 2014, 2015, and 2017. Therefore, the sampling frame was also changed since the number of urban areas [Municipalities] reached 217 in 2015 from 58 in 2011 and now reached 263, which consists of 59 percent of the national population. As per the Constitution of Nepal 2015, seven provinces/states have been established [16]. All provinces and almost all districts have rural and urban areas. The political map and terms for the rural area are changed as 'Rural Municipality' [*Gaupalika*] and urban areas as 'Municipality' or Metropolitan' [*Nagarpalika*] including its unit 'wards'. The average size of the ward is 104 and 800 households in rural and urban areas, respectively [15]. Altogether 11, 473 households were selected as sample size, but the information was obtained from 11,040 household heads.

## Sampling

The stratified sampling method was applied following two stages sampling in rural and three stages sampling in urban areas. The CBS had established an enumeration area [EA] in each ward. Households were selected from the primary sampling unit [PSU], where wards were selected as PSU in rural areas. Nevertheless, in rural areas, households were selected from EA, where an EA was selected from PSU. Fourteen sampling strata were made in each province covering the rural and urban areas. In the initial phase, 383 wards were selected with population proportional to size method. Simultaneously 30 households per cluster were selected, applying an equal probability systematic sampling method in the second phase. Details of the sampling procedure were obtained from the Nepal Demographic and Health Survey 2016, which is publicly available from the DHS program [15].

## Research tool

The NDHS was an extensive survey that included various individual and household-related information. NDHS 2016 used six different questionnaires that included 12 different health-related topics, including background characteristics of households. We extracted data from the household questionnaire. The questionnaire had included background and sociodemographic features of household and family members, household dwellings, ownership, and migration, etc. [15]. Proxy measures of household's dwelling and assets were assessed to categorize the wealth index.

The questionnaire was pretested in the different geographical locations of Nepal. Questions were translated into Nepali, Maithili, and Bhojpuri. Three different districts were selected for pretesting the questionnaire: Dhading for Nepali language, Sarlahi for Maithili, and Bara for Bhojpuri. After completing the pre-test, the involved team members shared their experiences during the pre-test in the debriefing session and modified the questionnaire [15].

## Data collection

Most of the enumerators and field staff had previous experience with fieldwork as well as NDHS. They received two weeks long main training, and one week of technical assessment training started on 15$^{th}$ May 2016. A total of 64 enumerators and 16 supervisors were appointed as per their performance. A total of 16 teams were formed, and each team had one supervisor, one male interviewer, and three female interviewers. These enumerators and field staff were deployed in Kathmandu valley first under the close supervision of trainers and quality controllers. After completing fieldwork in Kathmandu, they were called back to come together to share their experience where they shared their problems and queries. Following these events, all field staff was deployed to the respective districts. Data collection work was finished on 31$^{st}$ January 2017 [15].

## Data processing and analysis

Data were collected using computer devices named computer-assisted personal interviewing. All data were electronically transferred to the research organization (New ERA) office. Data were further inspected to remove the errors or inconsistencies and edited accordingly. The data entry and editing process was completed using CSPro software. Secondary data editing was completed in the second week, and in the last week of February 2017, the final data editing was completed by the DHS program [15]. We used the weighted sample from the survey.

In the study, respondents' and households' sociodemographic characteristics were considered independent variables, and knowledge about LF transmission was the dependent variable. Descriptive (frequencies and percentage), bivariate (association: chi-square test), and multivariate analyses (predictors: binary logistic regression) were performed to interpret the results. We developed two models for multivariate analysis. In the Model I, we observed the wealth status of respondents and its relation with knowledge about LF. In the Model II, we adjusted the sociodemographic characters of respondents along with wealth quintiles. We assessed the variables to determine the multi-collinearity among the variables before adjusting the variables in multivariate analysis (in Model II) and found no multi-collinearity issues among the variables. We used IBM SPSS Statistics version 26 to analyse the data.

## Ethical consideration

NDHS 2016 protocol was reviewed and approved by Nepal Health Research Council, Kathmandu, and IFC Institutional Review Board, Rockville, Maryland, USA. Consent was taken

before the interview. Information obtained from respondents was kept confidential with anonymity. The 'national ethical guidelines for health research in Nepal and standard operating procedures' were followed throughout the survey processes [15, 17].

## Results

### Descriptive statistics of the household head/respondents

More than two-thirds, 7581(69%) of respondents were male. Nearly one-fourth, 2562 (23%) were aged 35 to 44 years. Nearly four out of ten, 4259 (39%) were from rural areas. Forty-seven (n = 5134) and 46 percent (n = 5125) of the total respondents were from hill and Terai, respectively. Similarly, 2521(23%) of the respondents were from Bagmati province, and 18 percent each from province one and province two. Nineteen percent of the respondents belonged to middle wealth status.

Migration to urban from rural areas has been increasing gradually. Nearly 17 percent population lived in urban areas in 2011, which increased to 59 percent in 2016. Forty-seven percent (n = 5129) of the respondents' family members migrated within the past ten years. Three-fourth, 8290 (75%) of the respondents had mosquito nets in their homes. Twenty-nine (n = 3229) and 5294 (52%) of respondents had Radio and television at their homes, respectively. [Table 1]

### Knowledge of mode of transmission of Lymphatic Filariasis

An overwhelming majority, 10142 (92%) of the respondents had ever heard about LF locally called 'Hattipaile' (Elephantiasis: like an elephant's leg). However, just above a fourth, 3079 (28%) of the respondents had correct knowledge of its mode of transmission. Total three percent of respondents had erroneous information about the mode of transmission of LF that it transmits through contaminated foods or others. [Table 2].

### Association between respondents' characteristics and their knowledge of mode of transmission of Lymphatic Filariasis

Data showed that a significantly higher proportion of male respondents (n = 2222, 29%) than female respondents (n = 857, 25%) knew the mode of transmission of LF (p = 0.001). In regards to the age group, a higher proportion of respondents aged 15–24 years (n = 203, 33%) and 35–44 years (n = 809, 32%) had correct knowledge of modes of transmission of LF, and the association is statistically significant as well (p = 0.001). Three out of ten respondents (n = 2055) from urban areas knew the mode of transmission of LF compared to 24 percent (n = 1024) of rural respondents (p = 0.001). In the same way, 30 percent (n = 1560) of the respondents from hill knew the mode of transmission of LF, followed by Terai (n = 1368, 27%) and mountain (n = 151, 19%) (p = 0.001).

A significantly higher proportion of the respondents from Bagmati (n = 843, 33.4%) and Gandaki Province (389, 33.1%) knew the mode of transmission of LF compared to other provinces (p = 0.001). Data showed that knowledge of the mode of transmission of LF increased with an increase in wealth status. Forty-four percent (n = 999) of the respondents from the richest respondents knew the mode of transmission of LF compared to 16 percent (n = 346) of the poorest respondents (p = 0.001). However, migrated respondents had a low level of knowledge about LF. Nearly three out of ten (n = 1720, 29%) respondents who were not migrated within a decade of survey year knew the mode of transmission of LF compared to 27 percent (n = 1359) of those who were migrated.

**Table 1. Background characteristics of respondents (head of household).**

| Variables | Response Category | N | % |
|---|---|---|---|
| Sex of the head of household | Male | 7581 | 68.7 |
| | Female | 3459 | 31.3 |
| Age of household head | 15–24 | 625 | 5.7 |
| | 25–34 | 2240 | 20.3 |
| | 35–44 | 2562 | 23.2 |
| | 45–54 | 2358 | 21.4 |
| | 55–64 | 1810 | 16.4 |
| | 65 and above | 1445 | 13.1 |
| Place of residence | Urban | 6781 | 61.4 |
| | Rural | 4259 | 38.6 |
| Ecological zone | Hill | 5134 | 46.5 |
| | Mountain | 781 | 7.1 |
| | Terai | 5125 | 46.4 |
| Province | Bagmati | 2521 | 22.8 |
| | Province 1 | 2004 | 18.2 |
| | Province 2 | 2014 | 18.2 |
| | Gandaki | 1173 | 10.6 |
| | Lumbini | 1793 | 16.2 |
| | Karnali | 619 | 5.6 |
| | Sudur Paschim | 915 | 8.3 |
| Wealth status | Poorest | 2234 | 20.2 |
| | Poorer | 2225 | 20.2 |
| | Middle | 2065 | 18.7 |
| | Richer | 2240 | 20.3 |
| | Richest | 2276 | 20.6 |
| Migrated in the past 10 years | No | 5911 | 53.5 |
| | Yes | 5129 | 46.5 |
| Have mosquito net | No | 2750 | 24.9 |
| | Yes | 8290 | 75.1 |
| Has Radio in household | No | 7811 | 70.7 |
| | Yes | 3229 | 29.3 |
| Has a TV in the household | No | 5346 | 48.4 |
| | Yes | 5694 | 51.6 |
| Total | | 11040 | 100.0 |

Source: NDHS 2016

Thirty percent (n = 2463) of the respondents who had mosquito net knew the mode of transmission of LF compared to 22 percent (n = 616) of those who had no mosquito net (p = 0.001). Similarly, one-third of the respondents who had Radio and TV knew the mode of transmission of LF compared to those who do not have Radio (n = 2029, 26%) and TV (n = 1183, 22%) (p = 0.001) [Table 3].

## Multivariate analysis of respondents' characters and knowledge of the mode of transmission of Lymphatic Filariasis

We calculated the odds ratio (OR) and adjusted odds ratio (AOR) to predict the association among the variables. Findings showed that wealth status had a positive effect on knowledge of

**Table 2. Awareness of Lymphatic Filariasis (*Hattipaile*) and its mode of transmission.**

| Variables | Response | N | % |
|---|---|---|---|
| Heard about Lymphatic Filariasis (*Hattipaile*) | No | 898 | 8.1 |
| | Yes | 10142 | 91.9 |
| Lymphatic Filariasis transmit: | | | |
| *through the mosquito bite* | No | 7961 | 72.1 |
| | Yes | 3079 | 27.9 |
| *from contaminated food* | No | 10866 | 98.4 |
| | Yes | 174 | 1.6 |
| *a curse from god* | No | 11039 | 100.0 |
| | Yes | 1 | 0.0 |
| *others* | No | 10886 | 98.6 |
| | Yes | 154 | 1.4 |
| Total | | 11040 | 100.0 |

Source: NDHS 2016

LF. The Model I and II showed the same result that the respondents having the poorest wealth status appeared to be 77 to 78 percent less likely to know the mode of transmission of LF (AOR = 0.22, 95%CI: 0.18–0.27, p = 0.001) compared to the respondents with richest wealth status. Female respondents who were household heads had lower odds (AOR = 0.78, 95%CI: 0.70–0.86, p = 0.001) of knowing LF transmission compared to the male household head. Similarly, younger respondents had more knowledge about the mode of transmission of LF compared to elder respondents.

Interestingly, the respondents from Terai and mountain had lower knowledge of LF mode compared to the respondents from the hill. Similarly, the respondents who lived in Gandaki Province were more likely to have knowledge (AOR = 1.22, 95%CI: 1.04–1.42, p = 0.01) on LF modes compared to the respondents from Bagmati Province. Likewise, possession of mosquito nets and Radio were significant predictors for known modes of transmission of LF. The respondents who had a mosquito net and Radio were 29 percent more likely to know the mode of transmission of LF compared to the respondents who had no mosquito net and Radio (p = 0.001) [Table 4].

## Discussion

Sociodemographic characteristics of the respondents such as sex of household head; the age of the respondents, place of residence, ecological belt, provinces, wealth status, migration within the past ten years, and possession of mosquito net, Radio, and TV were significantly associated with the knowledge of the mode of transmission of LF. However, wealth status, sex, residence setting, ecological belt, and having a mosquito net, and Radio were the significant predictors for knowing the mode of transmission of LF. Similarly, age and Province also appeared as significant predictors for knowing the mode of transmission of LF to some extent. As per the objective, it can be concluded that the wealth status of the household was significantly associated with the knowledge of transmission of LF, moreover, wealth status was also appeared as a significant predictor for having knowledge of the mode of transmission of LF.

In this study, younger respondents (15–24 years) had comparatively more knowledge than elder ones(≥25 years); however, Samoa's findings were quite different from 87 percent of the elder respondents (≥25 years) had heard about LF compared to 67 percent of the respondent's age under 25 years [6].

**Table 3. Background characteristics of respondents and knowledge of mode of transmission of Lymphatic Filariasis.**

| Variables | Response Category | Knowledge of LF transmission | | Total | Chi square |
|---|---|---|---|---|---|
| | | No: n(%) | Yes: n(%) | | p value |
| Sex of the head of household | Male | 5358 (70.7) | 2222 (29.3) | 7581 | Chi-square 17.9, p = 0.001 |
| | Female | 2602 (75.2) | 857 (24.8) | 3459 | |
| Age of household head | 15–24 | 422 (67.5) | 203 (32.5) | 625 | Chi-square 80.5, p = 0.001 |
| | 25–34 | 1607 (71.7) | 633 (28.3) | 2240 | |
| | 35–44 | 1752 (68.4) | 809 (31.6) | 2562 | |
| | 45–54 | 1674 (71.0) | 683 (29.0) | 2358 | |
| | 55–64 | 1370 (75.7) | 440 (24.3) | 1810 | |
| | 65 and above | 1135 (78.5) | 310 (21.5) | 1445 | |
| Place of residence | Urban | 4726 (69.7) | 2055 (30.3) | 6781 | Chi-square 57.0, p = 0.001 |
| | Rural | 3235 (76.0) | 1024 (24.0) | 4259 | |
| Ecological zone | Hill | 3574 (69.6) | 1560 (30.4) | 5134 | Chi-square 60.6, p = 0.001 |
| | Mountain | 630 (80.7) | 151 (19.3) | 781 | |
| | Terai | 3756 (73.3) | 1368 (26.7) | 5125 | |
| Province | Bagmati | 1679 (66.6) | 843 (33.4) | 2521 | Chi-square 87.6, p = 0.001 |
| | Province 1 | 1441 (71.9) | 563 (28.1) | 2004 | |
| | Province 2 | 1500 (74.5) | 514 (25.5) | 2014 | |
| | Gandaki | 784 (66.9) | 389 (33.1) | 1173 | |
| | Lumbini | 1347 (75.1) | 446 (24.9) | 1793 | |
| | Karnali | 476 (76.9) | 143 (23.1) | 619 | |
| | Sudur Paschim | 734 (80.2) | 181 (19.8) | 915 | |
| Wealth status | Poorest | 1888 (84.5) | 346 (15.5) | 2234 | Chi-square 461.4, p = 0.001 |
| | Poorer | 1692 (76.1) | 533 (23.9) | 2225 | |
| | Middle | 1530 (74.1) | 534 (25.9) | 2065 | |
| | Richer | 1574 (70.2) | 667 (29.8) | 2240 | |
| | Richest | 1277 (56.1) | 999 (43.9) | 2276 | |
| Migrated in the past 10 years | No | 4192 (70.9) | 1720 (29.1) | 5911 | Chi-square 4.1, p = 0.043 |
| | Yes | 3769 (73.5) | 1359 (26.5) | 5129 | |
| Having mosquito net | No | 2134 (77.6) | 616 (22.4) | 2750 | Chi-square 75.6, p = 0.001 |
| | Yes | 5827 (70.3) | 2463 (29.7) | 8290 | |
| Having Radio in household | No | 5781 (74.0) | 2029 (26.0) | 7811 | Chi-square 67.7, p = 0.001 |
| | Yes | 2180 (67.5) | 1050 (32.5) | 3229 | |
| Having a TV in household | No | 4164 (77.9) | 1183 (22.1) | 5346 | Chi-square 182.0, p = 0.001 |
| | Yes | 3797 (66.7) | 1896 (33.3) | 5694 | |
| Total | | 7961 (72.1) | 3079 (27.9) | 11040 | |

Source: NDHS 2016

Note: n = number, % = percentage

A study conducted in Egypt shows that 59 percent of the research participant knew the mode of transmission of LF that was quite higher than this study. Similarly, males had more knowledge and a better understanding of LF than females [1], which is similar to the findings of our study. This study showed that females had lower odds ratio or 28 percent less likely to know the mode of transmission of LF. A cross-sectional study conducted in Nigeria shows that more than half (51%) of the participants had heard about LF, but just nine percent had good knowledge. Nearly one out of ten (9%) knew the mode of transmission and eight percent of the respondents perceived that the mosquitoes were related to it [7]. Similarly, another study

**Table 4. Adjusted odds ratio of having knowledge of transmission of Lymphatic Filariasis.**

| Predicators | Characters | Model I | | | | Model II | | | |
|---|---|---|---|---|---|---|---|---|---|
| | | OR | 95% CI | | | AOR | 95% CI | | |
| | | | Lower | Upper | p value | | Lower | Upper | p value |
| Wealth Status | Poorest | 0.234 | 0.204 | 0.270 | 0.001 | 0.223 | 0.183 | 0.272 | 0.001 |
| | Poorer | 0.402 | 0.354 | 0.457 | 0.001 | 0.388 | 0.331 | 0.454 | 0.001 |
| | Middle | 0.446 | 0.392 | 0.508 | 0.001 | 0.453 | 0.391 | 0.524 | 0.001 |
| | Richer | 0.542 | 0.479 | 0.612 | 0.001 | 0.533 | 0.468 | 0.608 | 0.001 |
| | Richest (ref.) | 1.00 | | | | 1.00 | | | |
| Sex of the head of household | Male (ref.) | | | | | 1.00 | | | |
| | Female | | | | | 0.776 | 0.702 | 0.859 | 0.001 |
| Age of household head | 15–24 (ref.) | | | | | 1.00 | | | |
| | 25–34 | | | | | 0.805 | 0.659 | 0.983 | 0.033 |
| | 35–44 | | | | | 0.915 | 0.751 | 1.116 | 0.382 |
| | 45–54 | | | | | 0.750 | 0.611 | 0.920 | 0.006 |
| | 55–64 | | | | | 0.625 | 0.504 | 0.775 | 0.001 |
| | 65 and above | | | | | 0.540 | 0.432 | 0.675 | 0.001 |
| Place of residence | Urban (ref.) | | | | | 1.00 | | | |
| | Rural | | | | | 1.115 | 1.009 | 1.233 | 0.033 |
| Ecological zone | Hill (ref.) | | | | | 1.00 | | | |
| | Mountain | | | | | 0.779 | 0.637 | 0.954 | 0.016 |
| | Terai | | | | | 0.718 | 0.626 | 0.823 | 0.001 |
| Province | Bagmati (ref.) | | | | | 1.00 | | | |
| | Province 1 | | | | | 1.117 | 0.957 | 1.304 | 0.161 |
| | Province 2 | | | | | 0.999 | 0.828 | 1.206 | 0.994 |
| | Gandaki | | | | | 1.218 | 1.041 | 1.424 | 0.014 |
| | Lumbini | | | | | 0.885 | 0.750 | 1.044 | 0.147 |
| | Karnali Province | | | | | 1.157 | 0.926 | 1.446 | 0.200 |
| | Sudur Paschim | | | | | 0.844 | 0.686 | 1.037 | 0.107 |
| Migrated in the past 10 years | No (ref.) | | | | | 1.00 | | | |
| | Yes | | | | | 1.026 | 0.935 | 1.126 | 0.587 |
| Having mosquito net | No (ref.) | | | | | 1.00 | | | |
| | Yes | | | | | 1.292 | 1.146 | 1.458 | 0.001 |
| Having Radio in household | No (ref.) | | | | | 1.00 | | | |
| | Yes | | | | | 1.290 | 1.173 | 1.418 | 0.001 |
| Having TV in the household | No (ref.) | | | | | 1.00 | | | |
| | Yes | | | | | 0.936 | 0.836 | 1.049 | 0.258 |
| Constant | | 0.782 | | | | 0.983 | | | |
| Cox & Snell R Square | | 0.043 | | | | 0.060 | | | |
| -2 Log likelihood | | 12586.34 | | | | 12384.57 | | | |

Source: NDHS 2016

from Brazil shows that 52, and 16, of the participants, expressed that LF is a disease of swelling legs and transmitted by the mosquito. These studies also support the results of this study [18].

It was noticed that the majority (51%) of the respondents knew the mode of transmission of LF. Ten and six percent expressed that they got information from Radio and TV respectively about LF in Nigeria [19]. Similarly, one-third (34%) of the respondents stated that the mode of

transmission of LF was mosquito bites, and the proportion was higher than our study, and three percent expressed that LF was caused by worms [20].

Various studies showed that knowledge about LF seemed low among community people in Nepal. A community-based cross-sectional study conducted in Kailali, Kapilvastu, and Dhading Districts of Nepal shows that near to half (48%) of the respondents expressed mosquito as a cause of LF, which was higher than this study [21]. However, 77 percent of the participants knew the mode of transmission of LF as the bite of mosquitoes, and 34 percent had used bed nets. The study conducted in Malaysia further shows females than males, aged more than 40 years, and low-income participants were more likely to have awareness; however, statistically not significant [22], which was quite controversial results with this study. Nevertheless, another study supports this results that male respondents than females and aged less than 40 had higher odds ratios of having LF-related knowledge but not statistically significant [23].

Other studies showed different results about socioeconomic status and its association with the knowledge of LF mode of transmission. A study from the Philippines showed that most of the respondents had high knowledge about the LF but misconceptions on causes and mode of transmission. The study further showed that sociodemographic features such as age, sex, and wealth status were not significantly associated with knowledge and attitude towards LF [24], which was a controversial result.

Another study from Bangladesh showed that one-fourth of the participants stated the right response of source of LF as the bite of mosquitoes, and surprisingly 58 percent expressed that they had no idea about it, which showed the awareness level of the people [3]. Similarly, 87 percent of the respondents were unaware of the mode of transmission of LF in the Republic of Guinea [25]. A vast majority (94%) of the study participants did not express the cause of LF in the right way in Ogun state, Nigeria [26]. Therefore, not only Nepal but also most people from other countries were also unaware of the causes, including the mode of transmission of LF.

Titaley et al., in 2018, observed that 44 percent of the respondents aged between 36 to 45 years had high-level knowledge about LF compared to 42 percent aged less than 36 and 36 percent aged more than 45 years. The results show that the middle age group was more aware of LF but we found age between 15–24 years had more knowledge than other age groups. Moreover, 44 percent of the females had a high level of knowledge of LF compared to 28 percent of the males [27]. The result also showed different findings compared to this study. Therefore, it can be concluded that sociodemographic factors varied on the influence of knowledge of the mode of transmission of LF throughout the place and time. It might be the cause of different study settings. However, most of the studies showed that the poorer people seemed more vulnerable to not having adequate information about the mode of transmission of LF compared to the richer ones.

## Limitations

The study mainly focused on knowledge of the mode of transmission of LF. Quantitative data were extracted from the datasets of NDHS 2016, and there was no qualitative information. The wealth index was categorized as per proxy measures of household assets as stated by the participants. Results of the study might be influenced by recall bias due to respondents' reported information. The cross-sectional design was used to analyse the quantitative data of NDHS 2016. Therefore, the causality of factors associated with knowledge of the mode of transmission of LF could not be established. Some important variables that were normally associated with the knowledge of LF transmission were missed due to the unavailability of these variables in the NDHS dataset.

## Conclusion

Despite various programs launched by the government to make people aware and to reduce the prevalence of LF, knowledge of modes of transmission of LF is still found poor. Variation in the prevalence of LF range from less than one to 39 percent in different districts, with an average of 13 percent. It was noticed that 72 percent of the household heads were still unknown about the mode of transmission of LF. Findings showed that wealth status, sex, place of residence, eco-belts, bed nets, and Radio were significant predictors for knowing the mode of transmission of LF. Moreover, the age of household heads and provinces they lived in were also predictors for knowing the mode of transmission of LF to some extent.

The study identified the groups of populations with lower knowledge of the mode of transmission of LF. So, based on the study's findings, it would be better to create knowledge-based intervention programs targeting the people who are more vulnerable since they do not have adequate information about the mode of transmission of LF. Moreover, awareness related programmes on LF should be focused on the people having low economic conditions.

## Acknowledgments

The authors would like to thank the MEASURES DHS/DHS program for providing access to the NDHSs dataset.

## Author Contributions

**Conceptualization:** Ramesh Adhikari, Devaraj Acharya, Aakriti Wagle.

**Data curation:** Devaraj Acharya.

**Formal analysis:** Devaraj Acharya, Aakriti Wagle.

**Investigation:** Aakriti Wagle.

**Methodology:** Devaraj Acharya, Aakriti Wagle.

**Software:** Ramesh Adhikari, Devaraj Acharya.

**Supervision:** Ramesh Adhikari.

**Writing – original draft:** Devaraj Acharya.

**Writing – review & editing:** Ramesh Adhikari, Devaraj Acharya, Aakriti Wagle.

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
