## [Decision Letter · Decision Letter 0]

27 Sep 2021

PGPH-D-21-00202

Knowledge on mode of transmission of lymphatic filariasis: Does household wealth status have an effect on knowledge on mode of transmission?

Dear Dr. Acharya,

Thank you for submitting your manuscript to PLOS Global Public Health. After careful consideration, we feel that it has merit but does not fully meet PLOS Global Public Health’s publication criteria as it currently stands. Therefore, we invite you to submit a revised version of the manuscript that addresses the points raised during the review process.

The reviewers have identified several areas where your manuscript could be improved, particularly in the discussion section. There are also a number of clarifications and suggested minor modifications which you should consider when revising your manuscript.  

We look forward to receiving your revised manuscript.

Kind regards,

Ruth Ashton, Ph.D.

Academic Editor

Journal Requirements:

1. Please note that your Data Availability Statement is currently missing  the DOI/accession number of each dataset OR a direct link to access each database. If your manuscript is accepted for publication, you will be asked to provide these details on a very short timeline. We therefore suggest that you provide this information now, though we will not hold up the peer review process if you are unable.

Reviewers' comments:

Reviewer's Responses to Questions

**Comments to the Author**

1. Does this manuscript meet PLOS Global Public Health’s publication criteria? Is the manuscript technically sound, and do the data support the conclusions? The manuscript must describe methodologically and ethically rigorous research with conclusions that are appropriately drawn based on the data presented.

Reviewer #1: Yes

Reviewer #2: Yes

2. Has the statistical analysis been performed appropriately and rigorously?

Reviewer #1: Yes

Reviewer #2: Yes

3. Have the authors made all data underlying the findings in their manuscript fully available (please refer to the Data Availability Statement at the start of the manuscript PDF file)?

Reviewer #1: Yes

Reviewer #2: Yes

4. Is the manuscript presented in an intelligible fashion and written in standard English?

Reviewer #1: Yes

Reviewer #2: No

5. Review Comments to the Author

Reviewer #1: Abstract

1: please remove abbreviations from abstract section.

2: methods and materials section please replace ‘’…..was the independent variable’’ to ‘’….were the independent variable’’.

3: result section. ‘’…Radio were significant predictors for knowing LF’’. Is it about knowing LF or knowledge on mode of transmission? Please clarify this sentence. Here also radio is one of the predictors. How? Do you mean availability of radio or radio itself? I think it needs amendment.

4: the authors should clarify what type of data they used. In line number 19, ‘’ Secondary data from Nepal Demographic and 20 Health Survey [NDHS] 201 6 were analyzed.’’. In another section I saw, there is interview process. The authors need to clarify which type of data they used for their research. Primary or secondary?

Methods and materials

1. Some of abbreviations are written first without described first. So, please check abbreviations and acronyms throughout the document whether it is written in full word first (eg. Line number 94, IFC??).

2. Line number 95-97. The authors said ‘’ some 97 methods have been presented here.’’ And other methodological part is published elsewhere. I did not agree with this concept. Even if it is published elsewhere, you need to discuss/describe every method you used in this research.

3. Line number 98: population and sample: The section is mostly about sampling procedure and study setting. The authors should clarify specifically who is the source and study population in this study.

4. Were you included LF patients in your study or general population?

5. I did not see operational definition for knowledge. Who is translated the questionnaire?

6. What did you use to categorize wealth index? Clarify it in the method section.

Result section

1. The result section is clearly written; however, the authors should add numbers before per cent. For example, line number 169 ‘More than two-thirds (69%) of respondents were male.’ Instead you should say ‘‘More than two-thirds, 7581 (69%) of respondents were male’’.

2. Line number 182 and 183 seems conclusion of your finding. So take it to conclusion section. In result section you have to write only the findings only. No need to conclude or discuss about the finding.

3. Table 2: ‘’ How does Lymphatic Filariasis transmit: others’’. Please indicate ‘others’ in the footnote of the table. The content of the table is copy paste of the questionnaire. For example: ‘’How does Lymphatic Filariasis transmit: through the mosquito bite.’’ . the authors should amend the table to make it more clear. I suggest the following format for table 2.

Variables

Response N %

Heard about LF No

Yes

Transmission ways of LF through the mosquito bite No

Yes

from contaminated food No

Yes

a curse from god No

Yes

Others* No

Yes

*indicate others

4. Table 3 is not self-explanatory. For example: number in No and Yes column are not clear. I mean is it percent or numbers. If it is percent, show us the % in the yes or no cell. In addition to the percent, the authors should add the number of participants in each yes or no column. Again in the table 3, ‘’ Knowledge on Lymphatic Filariasis transmit through a mosquito bite’’ should be replaced to ‘’ knowledge on LF transmission’’. I think knowledge on mode of transmission should be operationalized.

5. Line number 205-216: The narration part should be written before table and show us the source table at the end of each paragraph. Again here while writing the findings the authors should put the number and % together by separating with coma (). Do this for all result section.

6. Line number 221 and 222: change ‘’(aOR = 0.22[0.18 - 0.27], p<0.001)’’ to (AOR=0.22, 95%CI: 0.18-0.27, p=0.001). The authors should present the exact P-value rather than using less than or greater than signs. For some variables the AOR and 95%CI were not narrated. For example line number 226-227, only p-value was shown. The authors have to revise this section.

7. Table 4: please, can you add p-values for each predictor you have found in Model II? The number in the table looks like copy paste of the SPSS output. Amend the numbers in the table. It is good to say for example ‘0.234’ rather than saying ‘.234’. Model I output is written for only wealth status. For other variables the cells are empty. Can you clarify it?

Discussion

The discussion section is not well written. The authors did not discuss major findings in the manuscript. Even though the result is written in good manner, I am not happy with the discussion section. Please discuss pertinent findings in this section including your assumptions why the difference between your result and other studies happen. Is the difference because of population or sample size or geography of the study setting or other? In your study many predictors were significantly associated with knowledge on mode of LF transmission. But these factors should be discussed separately by comparing with other related researches. In addition, I have seen that two different factors were discussed in one paragraph. This is not correct.

Line number 292 and 293 ‘’ Both results showed different findings compared to 293 this study.’’ What is your finding?

Reviewer #2: The paper by Adhikari et al. reports the results of a large study assessing the transmission knowledge of lymphatic filariasis, an endemic disease in Nepal.

The study is based on a large cross-sectional study in a large population, using a well-executed sampling design. The number of population included allows for accurate estimates. However, while the sampling was very well detailed, the description of the sample size is missing.

The main result concerns the economic status of the family, represented by the head of the family. The results clearly show that wealthier people are better informed about the mode of transmission, with a dose response effect.

On the form, probably the English could be improved on the whole document. In addition, there are a lot of errors of editing... : type, no space (line 246), double space; references between two points (line 200), aOR is not defined before using the acronym (line 221).... so there is a lot of form work to be done.

While the discussion does present the equivalent results from other countries, it lacks a discussion of the usefulness of this result. How to explain this result in Nepal, what to do with this result in terms of public health policy. A short paragraph on this aspect should be added, which only lines 316-318.

6. PLOS authors have the option to publish the peer review history of their article (what does this mean?). If published, this will include your full peer review and any attached files.

**Do you want your identity to be public for this peer review?** For information about this choice, including consent withdrawal, please see our Privacy Policy.

Reviewer #1: **Yes: **Chuchu Churko

Reviewer #2: No

---

## [Decision Letter · Decision Letter 1]

27 Jul 2022

PGPH-D-21-00202R1

Knowledge on mode of transmission of Lymphatic Filariasis: Does household wealth status have an effect on knowledge on mode of transmission?

Dear Dr. Acharya,

Thank you for submitting your manuscript to PLOS Global Public Health. After careful consideration, we feel that it has merit but does not fully meet PLOS Global Public Health’s publication criteria as it currently stands. Therefore, we invite you to submit a revised version of the manuscript that addresses the points raised during the review process.

We look forward to receiving your revised manuscript.

Kind regards,

Guglielmo Campus, Ph.D DDS

Academic Editor

Journal Requirements:

1. Please update your online Competing Interests statement. If you have no competing interests to declare, please state: “The authors have declared that no competing interests exist.”

Additional Editor Comments (if provided):

Reviewers' comments:

Reviewer's Responses to Questions

**Comments to the Author**

1. If the authors have adequately addressed your comments raised in a previous round of review and you feel that this manuscript is now acceptable for publication, you may indicate that here to bypass the “Comments to the Author” section, enter your conflict of interest statement in the “Confidential to Editor” section, and submit your "Accept" recommendation.

Reviewer #1: All comments have been addressed

Reviewer #3: All comments have been addressed

2. Does this manuscript meet PLOS Global Public Health’s publication criteria? Is the manuscript technically sound, and do the data support the conclusions? The manuscript must describe methodologically and ethically rigorous research with conclusions that are appropriately drawn based on the data presented.

Reviewer #1: Yes

Reviewer #3: Yes

3. Has the statistical analysis been performed appropriately and rigorously?

Reviewer #1: Yes

Reviewer #3: Yes

4. Have the authors made all data underlying the findings in their manuscript fully available (please refer to the Data Availability Statement at the start of the manuscript PDF file)?

Reviewer #1: Yes

Reviewer #3: Yes

5. Is the manuscript presented in an intelligible fashion and written in standard English?

Reviewer #1: Yes

Reviewer #3: Yes

6. Review Comments to the Author

Reviewer #1: My concerns on the manuscript are adequately addressed.

Reviewer #3: The authors appropriately revised the paper based on the initial review.

The data analysis method could have elaborated the variable selection criteria used to determine the sociodemographic variables retained in the final multivariable model II.

Also, it could be interesting to know how potential collinearity issues were addressed, if any, especially for the variables: having TV vs. Radio, and Province vs. Place of residence.

7. PLOS authors have the option to publish the peer review history of their article (what does this mean?). If published, this will include your full peer review and any attached files.

**Do you want your identity to be public for this peer review?** For information about this choice, including consent withdrawal, please see our Privacy Policy.

Reviewer #1: **Yes: **Chuchu Churko

Reviewer #3: **Yes: **Ruxton Adebiyi

---

## [Decision Letter · Decision Letter 2]

8 Sep 2022

PGPH-D-21-00202R2

Knowledge on the mode of transmission of Lymphatic Filariasis: Does household wealth status have an effect on knowledge on the mode of transmission?

Dear Dr. Acharya,

Thank you for submitting your manuscript to PLOS Global Public Health. After careful consideration, we feel that it has merit but does not fully meet PLOS Global Public Health’s publication criteria as it currently stands. Therefore, we invite you to submit a revised version of the manuscript that addresses the points raised during the review process.

We look forward to receiving your revised manuscript.

Kind regards,

Nnodimele Onuigbo Atulomah, PhD

Academic Editor

Journal Requirements:

Additional Editor Comments (if provided):

This manuscript has received many rounds of reviews and recommendations for revision. Despite this, I have observed that the title would benefit from minor modification. Consider the title to read; "Sociodemographic Characteristics as predictors of Knowledge regarding mode of transmission of Lymphatic Filariasis among population of Nepal", since the outcome variable in the study is Knowledge regarding mode of transmission of Lymphatic Filariasis.

The abstract in the Result section in lines 26-27 should be: "Findings showed that only 28 percent of household-heads correctly identified the mode of transmission of Lymphatic Filariasis in the study."

English rendering in Lines 32-34 should be clearly expressed.

Reviewers' comments:

Reviewer's Responses to Questions

**Comments to the Author**

1. If the authors have adequately addressed your comments raised in a previous round of review and you feel that this manuscript is now acceptable for publication, you may indicate that here to bypass the “Comments to the Author” section, enter your conflict of interest statement in the “Confidential to Editor” section, and submit your "Accept" recommendation.

Reviewer #1: All comments have been addressed

Reviewer #3: All comments have been addressed

2. Does this manuscript meet PLOS Global Public Health’s publication criteria? Is the manuscript technically sound, and do the data support the conclusions? The manuscript must describe methodologically and ethically rigorous research with conclusions that are appropriately drawn based on the data presented.

Reviewer #1: Yes

Reviewer #3: Yes

3. Has the statistical analysis been performed appropriately and rigorously?

Reviewer #1: Yes

Reviewer #3: I don't know

4. Have the authors made all data underlying the findings in their manuscript fully available (please refer to the Data Availability Statement at the start of the manuscript PDF file)?

Reviewer #1: Yes

Reviewer #3: Yes

5. Is the manuscript presented in an intelligible fashion and written in standard English?

Reviewer #1: Yes

Reviewer #3: Yes

6. Review Comments to the Author

Reviewer #1: All my comments have addressed by the authors.

Reviewer #3: The authors have addressed all the reviewers' concerns from the previous rounds of reviews.

As much as I recommend the publication of this manuscript, I would like to authors to consider two things:

1. The 2016 Nepal Demographic and Health Survey (NDHS) used sampling weights, and like most population surreys, it allows the sample to be statistically representative of the population. So I do not know if the authors had a reason for leaving out sample weighting in their analyses. Are the reported prevalence weighted or unweighted?

2. Publications have reported the unreliability of p-value especially as a limitation when you have a large sample size, where majority of the bivariate analysis are 'forcefully' significant. Hence, authors should consider other statistical indicators used to support the p-value, such as change-in-estimate etc.

7. PLOS authors have the option to publish the peer review history of their article (what does this mean?). If published, this will include your full peer review and any attached files.

**Do you want your identity to be public for this peer review?** For information about this choice, including consent withdrawal, please see our Privacy Policy.

Reviewer #1: No

Reviewer #3: **Yes: **Ruxton Adebiyi

---

## [Editor Report · Decision Letter 3]

20 Sep 2022

Sociodemographic characteristics as predictors of knowledge regarding mode of transmission of Lymphatic Filariasis among population of Nepal

PGPH-D-21-00202R3

Dear Dr. Acharya,

We are pleased to inform you that your manuscript 'Sociodemographic characteristics as predictors of knowledge regarding mode of transmission of Lymphatic Filariasis among population of Nepal' has been provisionally accepted for publication in PLOS Global Public Health.

Best regards,

Nnodimele Onuigbo Atulomah, PhD

Academic Editor
